# The Transcription Factor BrNAC19 Acts as a Positive Regulator of the Heat Stress Response in Chinese Cabbage

Shuai Yuan [1], Xiaoping Yong [2,3], Yuxin Lu [1], Yuxin Lei [1], Weijian Li [1], Qiuli Shi [1] and Xiuhong Yao [1,*]

1   Solid-State Fermentation Resource Utilization Key Laboratory of Sichuan Province, Department of Agriculture Forestry and Food Engineering, Yibin University, Yibin 644000, China; yuanbio11@163.com (S.Y.); luyuxin50304@163.com (Y.L.); leiyuxin2024@163.com (Y.L.); liweijian2025@163.com (W.L.); 2023090001@yibinu.edu.cn (Q.S.)
2   Rice and Sorghum Research Institute, Sichuan Academy of Agricultural Sciences (Deyang Branch, Sichuan Academy of Agricultural Sciences), Deyang 618099, China; yongxp@scsaas.cn
3   Vegetable Germplasm Innovation and Variety Improvement Key Laboratory of Sichuan Province, Sichuan Province Engineering Technology Research Center of Vegetables, Sichuan Academy of Agricultural Sciences, Chengdu 610000, China
*   Correspondence: yaoxiuhong@yibinu.edu.cn

**Abstract:** The frequent occurrence of excessive heat events driven by global warming poses a great threat to plant growth and food security. To survive in heat stress (HS) environments, plants have evolved sophisticated response mechanisms, and the transcriptional network that controls the expression levels of HS-inducible genes serves as an essential component of this process. NAC (NAM, ATAF1/2, and CUC2) transcription factors (TFs) play key regulatory roles in the abiotic stress responses of plants. However, the functional roles of NAC TFs in the heat stress response of Chinese cabbage remain elusive. In the present study, we identified the *Brassica rapa* NAC family transcription factor *BrNAC19* as a close homologue of Arabidopsis *NAC019* and found that it serves as a positive regulator of the HS response. BrNAC19 displayed inducible gene expression in response to HS, and its subcellular localization showed that it was localized in the nucleus. Heterologous expression of *BrNAC19* significantly enhanced the heat tolerance of plants and reduced the accumulation of reactive oxygen species (ROS) under HS conditions. Furthermore, our results demonstrated that BrNAC19 directly targeted and promoted the expression of superoxide dismutase 1 (*CSD1*) and catalase 2 (*CAT2*), two antioxidant-enzyme coding genes in Chinese cabbage. Altogether, our results suggest that BrNAC19 enhances heat stress tolerance by positively regulating the expression of genes involved in the HS response and ROS scavenging and exhibits potential as a target gene in Chinese cabbage breeding to increase heat stress tolerance.

**Keywords:** heat stress; BrNAC19; transcriptional regulation; ROS scavenging

## 1. Introduction

The frequent occurrence of excessive heat events caused by global warming has become a global problem, seriously affecting crop production and food security [1]. Therefore, studying the molecular mechanisms of the heat stress response (HS) will provide theoretical guidance for molecular breeding and cultivation of heat-resistant plants and is also an important issue that needs to be addressed in response to HS [2]. As sessile organisms, plants have evolved complex and diverse systems to cope with HS. These heat stress responses are divided into two types, namely, basal thermotolerance and acquired thermotolerance. Basal thermotolerance reflects the natural capacity of the plant to defend itself against extreme heat attacks, while acquired thermotolerance (also called thermopriming) involves the ability to remember and form conditioned reflexes to avoid danger, which is actually an adaptive response that occurs after a priori exposure to sublethal HS [3].

Heat stress disrupts protein folding and denatures correctly folded proteins, which activates molecular chaperones such as heat shock proteins (HSPs) to assist in protein folding. During this process, heat shock transcription factors (HSFs) are required to bind to heat shock element sequences of HSP promoters to enhance the expression of HSPs [4]. Constitutively expressed HSFA1s, including *HSFA1a*, *HSFA1b*, and *HSFA1d*, are activated under HS conditions, which enables HSFA1s to activate the expression of *HSFA2* [5]. On the other hand, HS causes the accumulation of reactive oxygen species (ROS), which causes damage to plant tissues [6]. When high temperatures trigger the heat stress response of plants, plant antioxidants including superoxide dismutase (SOD), ascorbate peroxidase (APX), and catalase (CAT) are activated and are responsible for the scavenging of ROS [7].

Recent studies in *Arabidopsis thaliana* (L.) *Heynh* elucidated complex transcriptional networks during HS comprising many transcriptional regulators, and portions of transcription factors from different families jointly construct a transcriptional cascade in response to heat stress [5]. These transcription factor families include No apical meristem, *Arabidopsis thaliana*-activating factor 1/2, Cup-shaped cotyledon 2 (NAC), WRKY, MYB, basic region/leucine zipper motif (bZIP), basic-Helix-Loop-Helix (bHLH), ethylene response factors (ERF), dehydration-responsive element binding (DREB), and more [8,9]. As plant-specific transcriptional factors, NACs widely modulate essential aspects of plant function, including responses to HS [10]. Several NACs have been implicated in stress responses, and these genes are associated with enhanced tolerance in crop plants, such as rice, peach, pepper, and maize [11–14]. In the model plant Arabidopsis, AtNAC019 interacts with regulators of C-repeat binding factors (RCFs) and undergoes dephosphorylation by RCF2. Dephosphorylated NAC019 binds to the promoters of *HSFA1b*, *HSFA6b*, *HSFA7*, and *HSFC1* to positively regulate the heat stress response [15]. The other three NAC members, JUNG-BRUNNEN1 (JUB1), *Arabidopsis thaliana*-activating factor 1 (ATAF1), and NAC055, are involved in the heat stress memory response [16,17]. Therefore, precise modulation of NAC proteins is expected to enhance thermotolerance in plants. However, further investigation is required to elucidate the molecular mechanisms by which NAC regulates heat stress responses in crops and vegetables.

Chinese cabbage (*Brassica rapa* L. ssp. *pekinensis*), a member of the cruciferae family, is highly favored by consumers due to its nutritional value, high yield, and storage and transport resilience [18]. As a cool-season vegetable [19], it thrives at optimal temperatures ranging from 15 °C to 25 °C and exhibits sensitivity to elevated heat conditions. Particularly, temperatures exceeding 30 °C can adversely affect the quality of the vegetable [20,21]. Consequently, investigating the mechanisms of heat tolerance in Chinese cabbage is of significant importance to the agricultural industry.

In this study, we revealed a positive role for the Chinese cabbage NAC transcription factor *BrNAC19* in plant basal thermotolerance. Overexpression of *BrNAC19* in Arabidopsis enhanced plant thermotolerance and promoted the expression of HS-response-related genes. In addition, overexpression of *BrNAC19* decreased ROS accumulation under heat stress conditions. Correspondingly, BrNAC19 also directly bound to the promoter of Chinese cabbage antioxidant enzyme-coding genes for transcriptional activation, including *BrCSD1* and *BrCAT2*. Collectively, the findings from our study provide valuable information for genetic manipulation to cultivate heat-tolerant cabbage in future plant breeding programs.

## 2. Materials and Methods

### 2.1. Plant Materials and Growth Conditions

The *A. thaliana* plants used in this study were on a Col-0 ecotype background. *BrNAC19-OE* was obtained via *Agrobacterium tumefaciens*-mediated floral transformation. Homozygotes were screened for hygromycin B resistance in the T2 population. The Chinese cabbage species used in this study is *B. rapa* ssp. *Pekinensis*.

Arabidopsis and Chinese cabbage seeds were sterilized with 70% ethanol, sown on Murashige and Skoog medium (MS; Phyto Technology Laboratories, Lenexa, KS, USA)

supplemented with 1.5% sucrose and 0.8% agar, and incubated at 4 °C in the dark for 3 days. The seedlings were then grown on LD (16/8) at 22 °C under white light (100 µmol/m$^2$/s).

### 2.2. RNA Extraction and RT-qPCR

Total RNA was extracted from Chinese cabbage or Arabidopsis seedlings using a TaKaRa MiniBEST Universal RNA Extraction Kit (TaKaRa, Kyoto, Japan) according to the manufacturer's protocol. The concentration of RNA was measured by a microspectrophotometer (Thermo Scientific NanoDrop, Waltham, MA, USA), and 2 µg of RNA was reverse-transcribed using M-MLV Reverse Transcriptase (Invitrogen, Carlsbad, CA, USA). RT-qPCR was performed using a Hieff® qPCR SYBR Green Master Mix (YEASEN, Shanghai, China) with a CFX96 Real-Time PCR Detection System (Bio-Rad, Hercules, CA, USA). A two-step PCR amplification program was adopted, which included a holding stage (95 °C 30 s), a cycling stage (step1: 95 °C 30 sec, step2: 95 °C 30 s, number of cycles: 40), and a melt curve stage. Gene expression was measured in three independent biological replicates, and each biological replicate contained three technical replicates. Transcript expression levels were normalized to those of *ACTIN2* (AT3G18780) for Arabidopsis and *BrEF-1α* (Bra031602) for Chinese cabbage. The specific primer sequences are listed in Supplemental Table S1.

### 2.3. Phylogenetic Analysis and Sequence Alignment

The BrNAC19 protein sequence was downloaded from the Brassicaceae Database (BRAD), and the phylogenetic tree was constructed using MEGA 11 software with the neighbor-joining method and 1000 bootstrap replicates. The protein alignment was performed using DNAMAN (version 5.2.2) software.

### 2.4. Subcellular Localization

The coding sequence (CDS) of *BrNAC19* was amplified and cloned into the pCambia1302-GFP vector, then transformed into *A. tumefaciens* GV3101 and infiltrated into the leaves of *Nicotiana benthamiana*. After incubation for 2–3 d, GFP fluorescence was detected under a fluorescence microscope (Leica, Wetzlar, Germany) with an argon laser at an excitation wavelength of 488 nm using a 500–520 nm band-pass filter.

### 2.5. Thermotolerance Assay

The basal thermotolerance assay for Arabidopsis was performed as previously described [22]. Briefly, seedlings grown in a chamber for 7 days were subjected to 44 °C for 90 min in a constant-temperature incubator for HS triggering. After triggering, seedlings were transferred to normal growth conditions for 5 days for recovery.

### 2.6. Physiological Phenotype Assay

For measurement of chlorophyll contents, Arabidopsis seedlings were soaked in 80% acetone and placed in darkness for 16 h at 4 °C, then the absorbance of the supernatant was measured at 647 and 663 nm. Chlorophyll quantification was performed as previously described [23].

For electrolyte leakage detection, seedlings were harvested and rinsed twice with deionized water. After incubation at room temperature for 24 h, the conductivity was measured using an electroconductivity meter (DDSJ-308A; Lei-ci, Shanghai, China). Samples were then boiled at 100 °C for 15 min and shaken at room temperature for another 1 h before the electrical conductivity was measured again. The relative ion leakage was calculated based on the ratio of electrical conductivity before and after boiling.

Histochemical staining with DAB and NBT was performed as previously described [24]. Briefly, Arabidopsis seedlings were vacuum-infiltrated with NBT (0.5 mg/mL) solutions for 2 h or DAB (2 mg/mL) solutions for 8 h. Samples were then decolorized in ethanol (95%) on the heater until the leaves' blading faded. $H_2O_2$ and $O_2{}^-$ levels were analyzed by specific detection kits following the manufacturer's instructions (Beyotime Biotechnology Co., Ltd., Shanghai, China).

### 2.7. EMSA Assay

EMSA assays were performed as previously described [25]. The full-length CDS of *BrNAC19* was cloned into pGEX-4T-1 to express the glutathione S-transferase (GST)-BrNAC19 recombinant protein. GST-BrNAC19 was purified using glutathione agarose (Cat no. G4510; MERCK, Darmstadt, Germany). Purified GST-BrNAC19 was incubated with the DNA probe, and the mixture was reacted in binding buffer (25 mM HEPES-KOH, pH 8.0, 50 mM KCl, 1 mM dithiothreitol and 10% glycerol) for 30 min and then separated on a polyacrylamide gel. The probe and protein were detected using an EMSA Kit (Thermo Fisher, Waltham, MA, USA).

### 2.8. Y1H Assays

Y1H assays were performed as previously described [26]. The full-length CDS of *BrNAC19* was cloned into the PGAD-T7 vector, and the promoters of *BrCSD1* and *BrCAT2* were cloned into the pHIS2 vector. The constructs and empty vector controls were transformed in the yeast strain AH109, which were grown on SD/-Leu-Trp-His dropout plates for interaction analysis. 3-Amino-1,2,4-triazole (3-AT) was applied to inhibit self-activation.

### 2.9. Dual-Luciferase Assay

The promoters of *BrCSD1* and *BrCAT2* were amplified from Chinese cabbage DNA and cloned into the pGreenII 0800-LUC vector. The reporters and effectors were transformed into Agrobacterium and infiltrated into the leaves of *N. benthamiana*. Luciferase and Renilla activities were quantified using a Dual-Luciferase Reporter Assay System (Promega, Madison, WI, USA) and detected with a GloMax 20–20 luminometer (Promega).

Transformation of Cabbage protoplasts was performed as described in a recent report [27]. Briefly, true leaves of Chinese Cabbage were digested for 6 h in an enzyme solution containing 1.5% cellulose R-10, 0.4% macerozyme R-10, 20 mmol/L KCl, 20 mmol/L MES (pH 5.7), 0.6 mol/L mannitol, 10 mmol/L $CaCl_2$, and 0.1% bovine serum albumin (BSA). Then, W5 (154 mmol/L NaCl, 125 mmol/L $CaCl_2$, 5 mmol/L KCl, and 2 mmol/L MES; pH 5.7) solution was added. After filtration and precipitation, the protoplasts were resuspended in MMG solution (0.3 mol/L mannitol, 15 mmol/L $MgCl_2$, and 4 mmol/L MES, pH 5.7). Lastly, protoplasts were subjected to PEG-mediated transfection. After incubation for 24–36 h, Luciferase and Renilla activities were quantified.

### 2.10. Statistical Analysis

One-way analysis of variance (ANOVA), two-way ANOVA, Student's *t* test, and Tukey's post hoc test were performed using SPSS_26.0 software. The band intensity of Western blot bands was measured with ImageJ (version 1.51j8) software.

## 3. Results

### 3.1. BrNAC19 Plays a Positive Role in Plant Basal Thermotolerance

To identify the Chinese cabbage NAC transcription factor involved in HS, we analyzed the gene expression pattern of cabbage NACs under heat stress, especially homologous genes of NAC members which have been reported to regulate HS in other species (Figure S1). Among the NAC members we tested, *Bra018998* was identified as a candidate with highly inducible gene expression in response to high temperature shock, and especially was greatly induced in the leaves of Chinese cabbage, suggesting that *Bra018998* may play regulatory roles in mediating the HS response in cabbage leaves (Figure 1A). The phylogenetic tree of NAC proteins in rapeseed, Arabidopsis, Camelina sativa, rice, wheat, and maize revealed that the Bra018998 protein shares a close evolutionary relationship with rapeseed BanC06g05930D, Arabidopsis AtNAC019, and Camelina sativa CsNAC55. Despite the phylogenetic tree showing that Bra018998 is most closely related to BanC06g05930D, a protein homology search showed that Bra018998 shared the highest identity with AtNAC019, and therefore we named Bra018998 BrNAC19 (Figure 1B). Subcellular localization showed that the fluorescence signal

of BrNAC19 appeared in the nucleus, whether at room temperature or high temperature, suggesting its function as a transcription factor (Figure 1D).

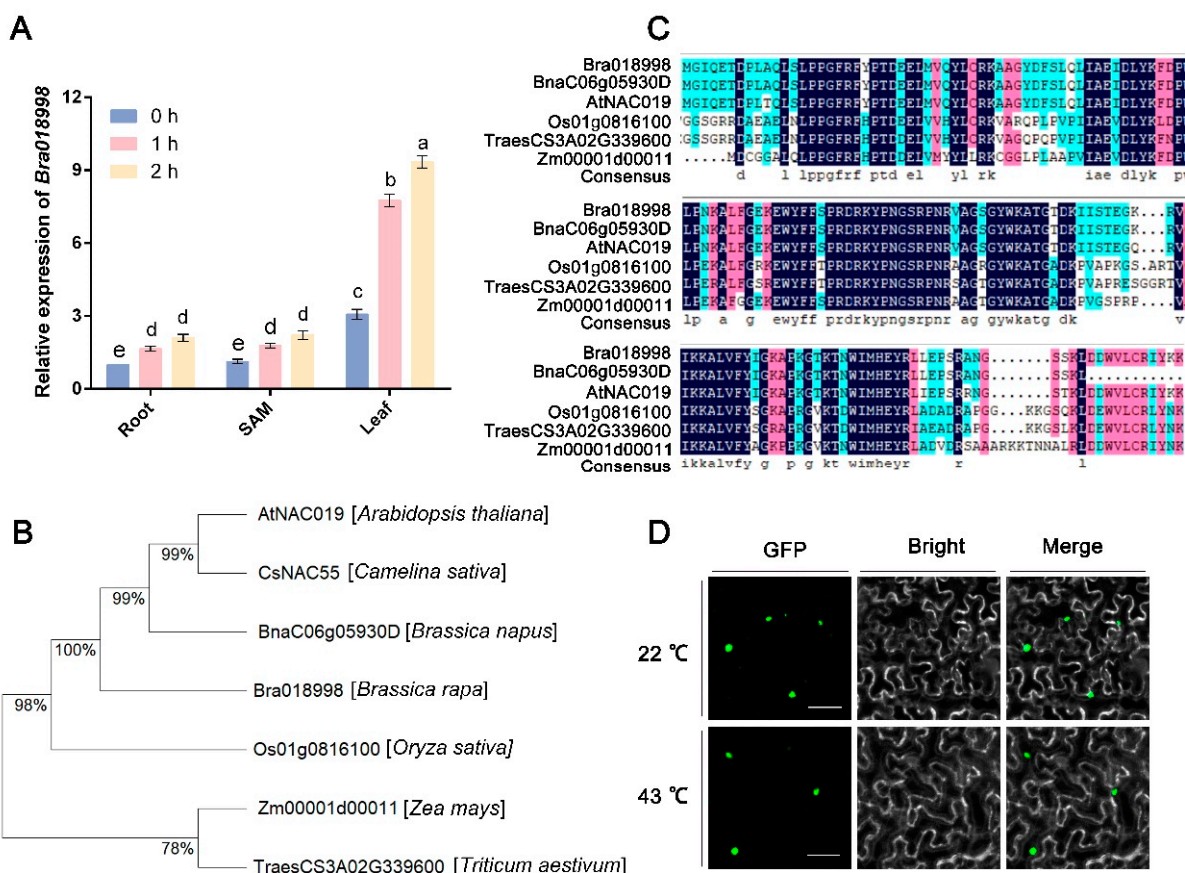

**Figure 1.** Heat-induced gene expression, homology analysis, and subcellular localization of *Bra018998*. (**A**) RT-qPCR detected the expression levels of *Bra018998* in the root, shoot apical meristems (SAM), and leaf of Chinese cabbage after high temperature exposure. Two-week-old Chinese cabbage seedlings grown under normal conditions were transferred to high temperature (43 °C) for the indicated time and then harvested for RNA extraction. The expression level of *Bra018998* in the roots at 0 h was set to one. Data are represented as the mean ± standard deviation (SD) of three biological replicates. (**B**) Phylogenetic analysis of Bra018998 with its orthologous genes based on their amino acid sequences. (**C**) Protein sequence multiple alignment of Bra018998 with its orthologous genes in other plant species. (**D**) Subcellular localization of Bra018998 in *N. benthamiana* leaf epidermis cells. Scale bars, 50 μm. The letters 'a' to 'e' above the bars indicate statistically significant differences between samples, and the presence of same letters between two groups indicates no significant differences (two-way ANOVA with Tukey's post hoc test; $p < 0.05$).

In Arabidopsis, NAC019 binds to the promoters of *HSFA1b*, *HSFA6b*, *HSFA7a*, and *HSFC1* for transcriptional activation, thus enhancing the thermotolerance of plants [15]. Since studies on NACs in crops have further confirmed the findings in Arabidopsis and supported the conserved function of stress response regulation [28,29], we speculated that BrNAC19 regulates heat tolerance similarly to AtNAC019. To explore its biological function, BrNAC19 was cloned and transformed in Arabidopsis under the control of the 35S promoter to generate the overexpression transgenic plants *BrNAC19-OE* (Figure S2). We employed a basal thermotolerance assay to elucidate the role of BrNAC19 in the heat stress response of Arabidopsis. First, 7-day-old seedlings were subjected to 44 °C for 90 min in a constant-temperature incubator and then allowed to recover in a growth chamber at 22 °C for 5 days before the survival rate of the plants was assessed. Seven-day-old *BrNAC19-OE* showed a similar phenotype to wild type Col-0 under normal conditions

(Figure 2A). However, after heat stress treatment and recovery, *BrNAC19-OE* exhibited enhanced thermotolerance, accompanied by an elevated survival rate, increased total chlorophyll content, and decreased electrolyte leakage (Figure 2B–D). Together, these results indicate that BrNAC19 positively confers basal thermotolerance in plants.

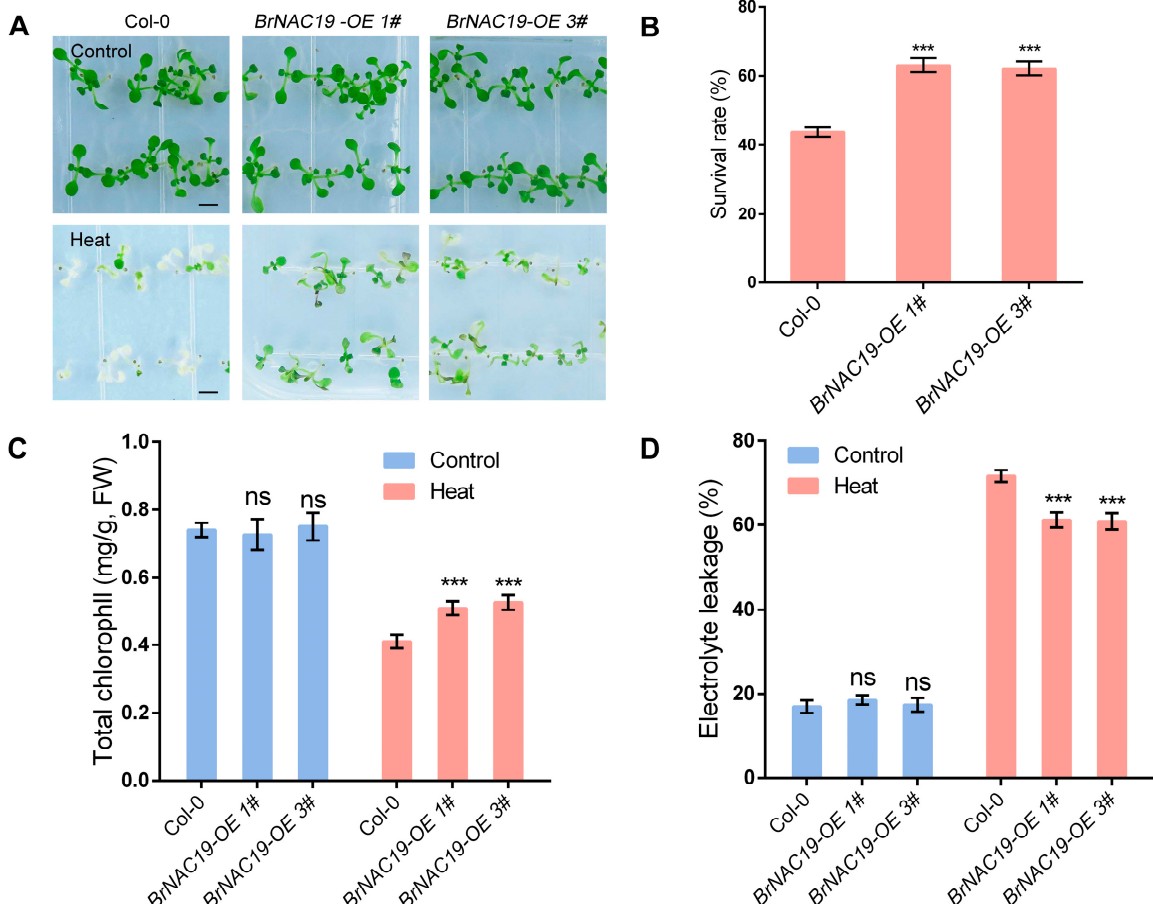

**Figure 2.** Overexpression of *BrNAC19* enhances plants thermotolerance in Arabidopsis. (**A**) Phenotypes of wild type (Col-0) and *BrNAC19-OE* seedlings in the basal thermotolerance assay (43 °C for 22 min and recovery at 22 °C for 5 days). The scale bar indicates 2 mm. 1# and 3# represent the numbering of different transgenic lines. (**B**) Survival rates of Col-0 and *BrNAC19-OE* seedlings in the basal thermotolerance assay. (**C**) Chlorophyll contents of the seedlings indicated in (**A**). (**D**) Electrolyte leakage assay of the seedlings indicated in (**A**). Data are presented as the mean ± standard deviation (SD) of three biological replicates. Significant differences compared with the wild type at same condition are noted (student's *t*-test, *** $p < 0.001$; ns indicates no significance).

### 3.2. BrNAC19 Activates HS-Response Gene Expression

Next, we investigated whether BrNAC19 modulates thermotolerance by transcriptionally regulating the key factors involved in the heat stress response. The HS response genes *CSD1/2*, *CAT1/2*, *HSF3*, and *HSFA1d* were selected for gene expression analysis in the wild type and overexpression lines of BrNAC19 [5,30,31]. Levels of these genes were upregulated in *BrNAC19-OE* plants after heat shock treatment (Figure 3A–F), indicating that BrNAC19 induces these genes to positively mediate the heat stress response.

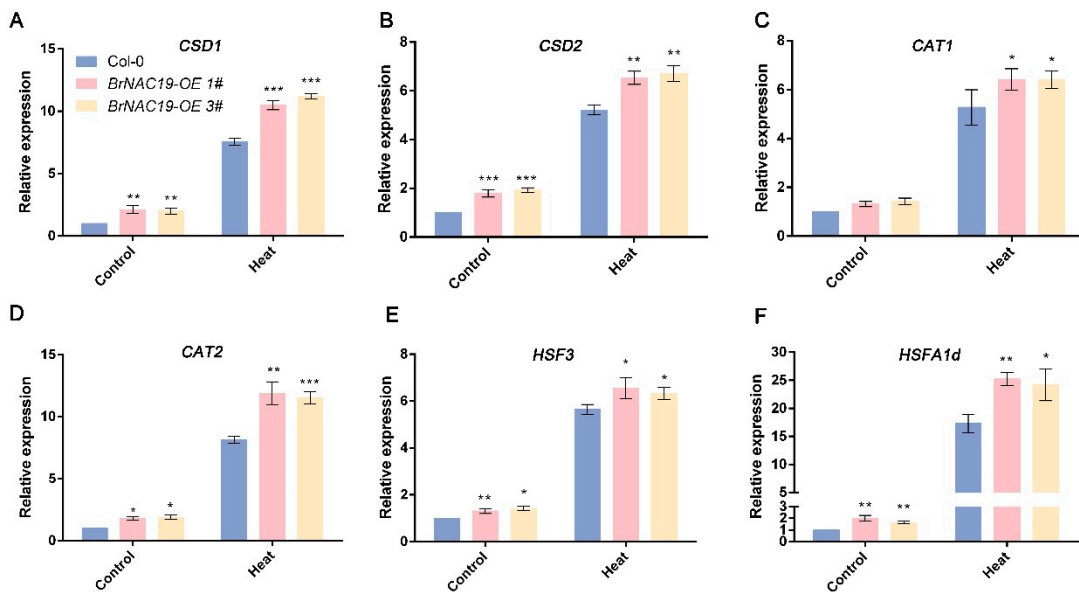

**Figure 3.** BrNAC19 promotes the expression of key regulators of HS response. (**A**–**F**) RT-qPCR detected the expression levels of *CSD1, CSD2, CAT1, CAT2, HSF3,* and *HSFA1d* in the indicated genotypes under different conditions. The expression level of each gene in Col-0 at 22 °C was set to one. All data are presented as means ± SD from three biological replicates. Significant differences compared with the wild type under the same conditions are noted (student's *t*-test, * $p < 0.05$, ** $p < 0.01$, *** $p < 0.001$).

### 3.3. Overexpression of BrNAC19 Decreases ROS Accumulation Under Heat Stress Conditions

Given that BrNAC19 activates the expression of transcription levels of genes encoding ROS-scavenging enzymes (Figure 3A–D), we supposed that BrNAC19-mediated heat stress tolerance was closely related to ROS scavenging. The production of ROS such as $H_2O_2$ and the superoxide anion ($O_2^-$) increases under heat stress [31], so we investigated the levels of $H_2O_2$ and $O_2^-$ via diaminobenzidine (DAB) staining and nitroblue tetrazolium (NBT) staining. Under normal conditions, the accumulation of $H_2O_2$ and $O_2^-$ in the wild type plants was only slightly higher than in the *BrNAC19-OE* plants. Conversely, significantly weakened staining was detected in *BrNAC19-OE* compared with WT under HS conditions (Figure 4A,B). To verify the histochemical staining, we quantified the accumulation of $H_2O_2$ and $O_2^-$ in the wild type and *BrNAC19-OE* lines. Consistent with the histochemical staining results, $H_2O_2$ and $O_2^-$ exhibited lower accumulation in the *BrNAC19-OE* plants (Figure 4C,D).

### 3.4. BrNAC19 Targets BrCSD1 and BrCAT2 for Activation

Since BrNAC19 promotes the expression of *CSD1/2* and *CAT1/2*, as well as decreasing the accumulation of ROS, in Arabidopsis under HS, we hypothesized that BrNAC19 may also play a role in $H_2O_2$ and $O_2^-$ scavenging by transcriptionally regulated genes encoding ROS-scavenging enzymes in Chinese cabbage. Analysis of the promoter sequences of *BrCSD1/2* and *BrCAT1/2* revealed that the NAC-binding site (TNCGTG/A) to which NAC members bind was present 1000 bp upstream of the translation initiation site of *BrCSD1* and *BrCAT2* (Figure 5A). Therefore, we hypothesized that BrNAC19 activates *BrCSD1* and *BrCAT2* by interacting with its promoters, at least partially. Electrophoretic mobility shifts assay (EMSA) revealed that BrNAC19 bound to the fragments of the *BrCSD1* and *BrCAT2* promoters containing a binding site, whereas the binding was abolished when mutant probes were used (Figure 5B). Furthermore, we tested whether BrNAC19 activates *BrCSD1* and *BrCAT2* transcription via a yeast-one hybrid assay, and the results indicated that BrNAC19 directly interacted with the promoters of *BrCSD1* and *BrCAT2* (Figure 5C). Lastly, we performed transient expression in tobacco leaves to analyze direct transcriptional activation of *BrCSD1* and *BrCAT2* by BrNAC19 using a dual-luciferase experiment. When

tobacco leaves were at 22 °C, BrNAC19 only mildly activated these two genes. However, when tobacco leaves were subjected to 43 °C heat shock, BrNAC19 strongly induced the expression of these two genes, indicating that BrNAC19 promotes the expression of *BrCSD1* and *BrCAT2* under thermal stimulation (Figure S3A,B). We further confirmed the transcriptional activation of *BrCSD1* and *BrCAT2* by BrNAC19 in Chinese cabbage protoplasts. When co-transfecting the reporter vector and BrNAC19-GFP, transcription of *BrCSD1* and *BrCAT2* was greatly induced (Figure 5D). In addition, we also detected the expression of *BrCSD1* and *BrCAT2* in Chinese cabbage, and the results indicated that the expression levels of *BrCSD1* and *BrCAT2* increased after heat treatment (Figure S4). All together, these results suggest that BrNAC19 directly targets *BrCSD1* and *BrCAT2* for transcriptional activation under heat stress.

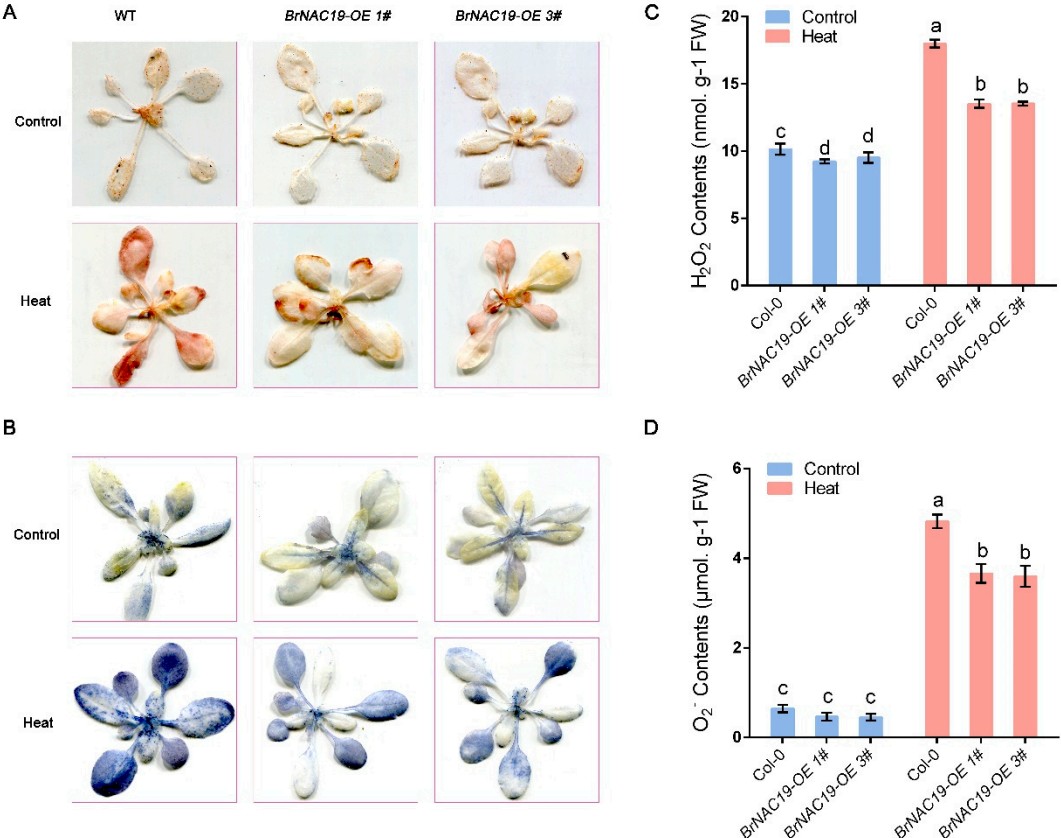

**Figure 4.** *BrNAC19-OE* rescues the ROS accumulation caused by HS. (**A**,**B**) Histochemical analysis of the generation of $H_2O_2$ and $O_2^-$ by staining with DAB and NBT in WT and *BrNAC19-OE* plants. Brown precipitation and blue spots represent the presence of $H_2O_2$ (**A**) and $O_2^-$ (**B**), respectively. (**C**,**D**) The levels of $H_2O_2$ (**C**) and $O_2^-$ (**D**) in WT and *BrNAC19-OE* plants. Data are presented as the mean ± standard deviation (SD) of three biological replicates. The letters 'a' to 'd' above the bars indicate statistically significant differences between samples, and the presence of the same letters between two groups indicates no significant differences (two-way ANOVA with Tukey's post hoc test; $p < 0.05$).

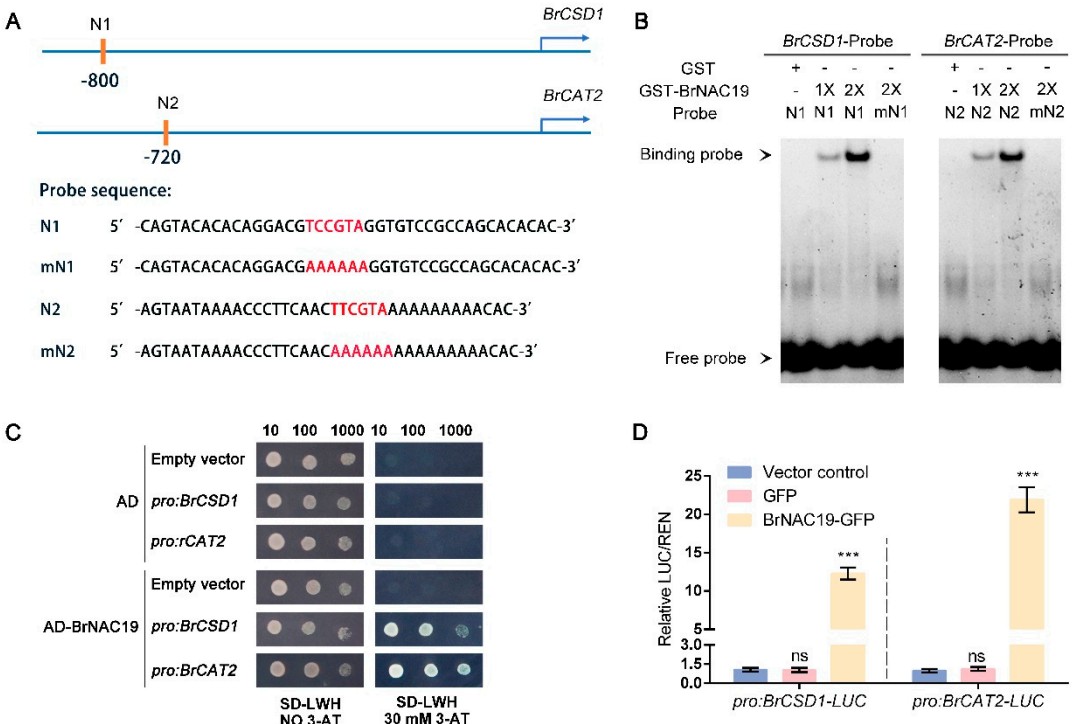

**Figure 5.** BrNAC19 directly induces the expression of *BrCSD1* and *BrCAT2*. (**A**) Schematic diagram of the binding sites in the *BrCSD1* and Br*CAT2* promoters, with the electrophoretic mobility shift assay (EMSA) probe sequences shown below the diagram. Red letters represent the NAC-binding site and mutation site. (**B**) EMSA revealed that BrNAC19 bound to the NAC-binding sites of the *BrCSD1* and Br*CAT2* promoters. The notation 2X indicates a twofold amount of glutathione S-transferase (GST)-BrNAC19 protein, and the probe sequence is shown in panel A. (**C**) Yeast one-hybrid (Y1H) assay revealed that BrNAC19 activates the *BrCSD1* and Br*CAT2* promoters. (**D**) Dual-luciferase assays indicated that BrNAC19 positively modulates transcription of *BrCSD1* and *BrCAT2* after heat shock. Data are presented as the mean ± standard deviation (SD) of three biological replicates (student's *t*-test, *** $p < 0.001$; ns indicates no significance).

## 4. Discussion

Heat shock poses a great threat to the growth, development, and production of crops and vegetables, thereby escalating as a serious challenge for food security as global warming progresses [32]. Plants enhance their tolerance through endogenous regulatory substances and heat-responsive transcriptional cascades [5,33]. In recent years, studies have elucidated the complex transcriptional regulatory networks involved in heat stress [5]. As plant-specific transcription factors, NAC family members play an important role in regulating plant growth, development, and stress response, including the heat stress response in multiple species [28]. In maize, a NAC transcription factor designated ZmNAC074 was identified that confers heat stress tolerance in transgenic Arabidopsis [14]. In cowpeas, overexpression of *VuNAC1/2* obviously enhance tolerance to drought, salinity, cold, and heat stress [34]. Upon analyzing the expression patterns of peach *NAC56* under abiotic stress, it was found that PpNAC56 responds to high-temperature stress, and overexpression of the *PpNAC56* gene in tomato plants conferred elevated heat tolerance [12]. *ONAC127* and *ONAC129* are responsive to heat stress and involved in the grain filling process of rice [35]. Using the CRISPR-Cas9 system, knockout of the *OsNAC006* transcription factor caused drought and heat sensitivity [36]. The membrane-associated NAC transcription factor OsNTL3 is involved in thermotolerance in rice, and inducible expression of the truncated form of *OsNTL3* without the transmembrane domain increases heat tolerance in rice seedlings [11]. In Arabidopsis, ANAC019 positively regulates plant heat stress response by activating HSFs [15]. However, it is still unclear how NAC transcription factors

regulate the heat stress response in Chinese cabbage. In this study, we found that BrNAC19 responds to heat stimuli and positively modulates the heat stress response by promoting the expression of HSFs and genes encoding ROS scavengers (Figures 1–3). Additionally, we identified *BrCSD1* and *BrCAT2* as direct targets of BrNAC19 (Figure 5), suggesting that Chinese cabbage enhances thermotolerance through BrNAC19-enhanced antioxidant enzyme synthesis. Further genetic analysis in Chinese cabbage will more thoroughly reveal the role that NAC transcription factors played in regulating heat stress tolerance in Chinese cabbage [37].

ROS typically act as major signaling molecules in the stress response of plants; among them, $H_2O_2$ and $O_2^-$ usually participate in the response to abiotic stresses such as heat stress to facilitate the transmission of stress signals [38]. However, the excessive accumulation of ROS causes lipid peroxidation, DNA damage, protein oxidation, and cell apoptosis, inflicting severe harm upon plants [7]. To eliminate the damage caused by ROS, plants have evolved an antioxidant defense system including diverse antioxidants like APX, SOD, and CAT [39]. To date, how ROS accumulation is regulated in Chinese cabbage during heat stress responses has remained poorly studied. In this study, we discovered that BrNAC19 promotes the expression of *CSD1* and *CAT2* both in Arabidopsis and Chinese cabbage, offering valuable insights into the mechanism by which Chinese cabbage utilizes NACs to regulate its adaptation to heat stress (Figures 3–5). Further investigation is needed to elucidate whether BrNAC19 contributes to the ROS scavenging system in Chinese cabbage under heat stress. In addition, whether BrNAC19 enhances antioxidant enzyme activity remains to be addressed.

## 5. Conclusions

Our results suggest that BrNAC19 plays a positive role in conferring thermotolerance in plants. When plants encounter high temperature stress, BrNAC19 induces the transcription of heat stress responsive genes and antioxidant enzyme-coding genes, thereby enhancing heat response intensity and preventing excessive accumulation of ROS (Figures 2–4). Importantly, our further molecular studies revealed the BrNAC19 targeted *BrCSD1* and *BrCAT2* for transcriptional activation (Figures 5 and S3). Overall, our study suggests that BrNAC19 induces the expression of heat stress response genes and ROS scavenging genes to enhance plant thermotolerance. Our data will help to develop detailed models of heat stress responses and clarify the roles that NAC transcription factors play in plants, which will facilitate the design of more heat stress-resilient crops and vegetables in response to global warming.

**Supplementary Materials:** The following supporting information can be downloaded at: https://www.mdpi.com/article/10.3390/horticulturae10121236/s1. Figure S1: RT-qPCR detected the expression of partial NACs tested under normal and high temperature conditions. Figure S2: Molecular characterization of *BrNAC19-OE* transgenic seedlings. Figure S3: BrNAC19 activated *BrCSD1* and *BrCAT2* transcription under heat. Figure S4: Heat treatment induced the expression of *BrCSD1* and *BrCAT2* in Chinese cabbage. Table S1. Primers used for RT-qPCR.

**Author Contributions:** Conceptualization, X.Y. (Xiuhong Yao); methodology, S.Y.; software, S.Y. and W.L.; validation, X.Y. (Xiaoping Yong); formal analysis, S.Y., X.Y. (Xiaoping Yong), and Y.L. (Yuxin Lei); resources, Y.L. (Yuxin Lu) and W.L.; data curation, Y.L. (Yuxin Lei); writing-original draft, X.Y. (Xiuhong Yao); writing-review and editing, Y.L. (Yuxin Lu); visualization, Y.L. (Yuxin Lu); supervision, S.Y.; project administration, X.Y. (Xiuhong Yao); funding acquisition, Q.S. and X.Y. (Xiuhong Yao). All authors have read and agreed to the published version of the manuscript.

**Funding:** This research was supported by the Joint Funds of the National Natural Science Foundation of China (Grant No. U22A20494), the Solid-State Fermentation Resource Utilization Key Laboratory of Sichuan Province (grant no. 2023GTZD03 to X.Y. and 2023GTZD05 to Q.S.), the Doctoral Research Launch Project of Sichuan Academy of Agricultural Sciences (NKYRCZX2024036), and the Project of Sichuan Province Engineering Technology Research Center of Vegetables (2023PZSC0303).

**Data Availability Statement:** The original contributions presented in the study are included in the article/supplementary material, and no other data were created.

**Conflicts of Interest:** The authors declare no conflict of interest.

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
