# Peer review of "The Transcription Factor BrNAC19 Acts as a Positive Regulator of the Heat Stress Response in Chinese Cabbage"

_horticulturae, doi:10.3390/horticulturae10121236_

Round 1
Reviewer 1 Report
Comments and Suggestions for Authors
This paper reports the identification of the BrNAC019 gene in Chinese cabbage and the generation of overexpression lines in Arabidopsis to investigate the gene's function under heat stress conditions. The BrNAC019 overexpression lines showed better tolerance to heat stress, reduced ROS production, and increased expression of heat inducible genes. Additionally, it was found that BrNAC019 in Chinese cabbage binds to the promoters of predicted downstream genes and enhances their expression under high-temperature conditions.
Considering the ongoing global warming and the gradual rise in temperatures, understanding the heat stress mechanisms in Chinese cabbage is critical for breeding heat-tolerant varieties. This study contributes to our understanding of the heat stress response in Chinese cabbage.
With some revisions, I believe the paper can be accepted.
1. It would be helpful to check if the expression levels of BrCSD1 and BrCAT2 also increase under heat treatment in Chinese cabbage.
Comments on the Quality of English Language1. English editing is necessary (e.g., capitalization, singular/plural usage
Author Response
Reviewer 1:
This paper reports the identification of the BrNAC019 gene in Chinese cabbage and the generation of overexpression lines in Arabidopsis to investigate the gene's function under heat stress conditions. The BrNAC019 overexpression lines showed better tolerance to heat stress, reduced ROS production, and increased expression of heat inducible genes. Additionally, it was found that BrNAC019 in Chinese cabbage binds to the promoters of predicted downstream genes and enhances their expression under high-temperature conditions.
Considering the ongoing global warming and the gradual rise in temperatures, understanding the heat stress mechanisms in Chinese cabbage is critical for breeding heat-tolerant varieties. This study contributes to our understanding of the heat stress response in Chinese cabbage.
With some revisions, I believe the paper can be accepted.
- It would be helpful to check if the expression levels of BrCSD1 and BrCAT2 also increase under heat treatment in Chinese cabbage.
Response: Thanks for the suggestion. We have detected the expression of BrCSD1 and BrCAT2 in Chinese cabbage, and the results indicated that the expression levels of BrCSD1 and BrCAT2 increased after heat treatment. Please see Figure S4 and lines 297-300, and line 373.
Comments on the Quality of English Language
- English editing is necessary (e.g., capitalization, singular/plural usage)
Response: Thank you for pointing out the deficiency in our manuscript. We have carefully checked the manuscript and revised it. The revisions could be tracked in the revised manuscript.
Reviewer 2 Report
Comments and Suggestions for Authors
Dear authors,
Hope you are doing well.
The manuscript on a transcription factor (BrNAC19) involved in high temperature tolerance in Chinese cabbage is interesting. This is because in the context of climate change, biotechnology and molecular biology are essential tools to face these challenges.
However, I consider that this manuscript shows serious flaws about methodoloy and associated results. Next, I indicate my main concerns:
Major concerns
1. Line 101. ‘approximately 2 ug’. It is important that all samples used in relative quantification analyzes by RT-qPCR are performed using the same amount of RNA. This guarantees obtaining accurate and reliable results. Otherwise, the expression levels may be variable and we will not be able to know if what is observed corresponds to a treatment effect or the effect of using more or less RNA. If you have used different concentrations of RNA (as the word roughly says), from my point of view, the RT-qPCR results are not reliable.
2. Additional information about RT-qPCR conditions are necessary. For example, temperatures, times, number of cycles, number of biological replicas and techniques used, etc.
3. There is no information on statistical analyses.
4. Tolerance vs resistance. I think that these concepts are different, however, you are using as synonyms. From my point of view, tolerance implies associated symptoms, but not resistance, among other differences. Furthermore, the molecular defense mechanisms against high temperatures are numerous and cannot be addressed by a single transcription factor. I think that you should use the ‘tolerance’ term through the text. Another possibility is that you justify with the literature why this gene is important for resistance to high temperature. However, I think you can't use both terms synonymously.
Minor concerns
1. Title. I suggest a change in the title: BrNAC19 transcription factor acts as a positive regulator of heat stress response in Chinese cabbage
2. Gene names should be written in italics. For example: BrNAC19 (lines 21-22), Superoxide Dismutase 1 (CSD1), Catalase 2 (CAT2),… Please, check the rest of the text.
3. Species names should be written in italics and should include the author name for the first citation. For instance: Arabidopsis thaliana (L.) Heynh. (line 55).
4. Please, review if this is correct: CUC2 (NAC) (line 58).
5. The second and subsequent times, the species names must be written with the abbreviated genus and species. For example, A. thaliana (line 90). Please check the rest of the text.
6. The name of the manufacturer must be accompanied by the country of manufacture. For example, lines 100, 103,…
Author Response
Reviewer 2:
Dear authors,
Hope you are doing well.
The manuscript on a transcription factor (BrNAC19) involved in high temperature tolerance in Chinese cabbage is interesting. This is because in the context of climate change, biotechnology and molecular biology are essential tools to face these challenges.
However, I consider that this manuscript shows serious flaws about methodoloy and associated results. Next, I indicate my main concerns:
Major concerns
- Line 101. ‘approximately 2 ug’. It is important that all samples used in relative quantification analyzes by RT-qPCR are performed using the same amount of RNA. This guarantees obtaining accurate and reliable results. Otherwise, the expression levels may be variable and we will not be able to know if what is observed corresponds to a treatment effect or the effect of using more or less RNA. If you have used different concentrations of RNA (as the word roughly says), from my point of view, the RT-qPCR results are not reliable.
Response: The concentration of RNA was measured by a micro spectrophotometer (Thermo Scientific NanoDrop, America). To ensure the same amount of RNA was used, different volumes of RNA were added based on the concentration but the total volume of the reaction is consistent when reverse transcription was performed. Also, we have rephrased this roughly description, please see lines 108-109.
- Additional information about RT-qPCR conditions are necessary. For example, temperatures, times, number of cycles, number of biological replicas and techniques used, etc.
Response: we have added this information. Namely, two step PCR amplification program was adopted, which contained a Holding Stage (95℃ 30 sec), a Cycling Stage (step1: 95℃ 30 sec, step2: 95℃ 30 sec, number of cycles: 40), and a Melt Curve Stage. Gene expression was measured in three independent biological replicates, each biological replicate contained three technical replicates. Please see lines 112-116.
- There is no information on statistical analyses.
Response: We have added a description about statistical analysis in the methods section. Please see lines 181-184. Some missing descriptions have also been added to the figure legends, please see lines 226-227, and lines 270-271.
- Tolerance vs resistance. I think that these concepts are different, however, you are using as synonyms. From my point of view, tolerance implies associated symptoms, but not resistance, among other differences. Furthermore, the molecular defense mechanisms against high temperatures are numerous and cannot be addressed by a single transcription factor. I think that you should use the ‘tolerance’ term through the text. Another possibility is that you justify with the literature why this gene is important for resistance to high temperature. However, I think you can't use both terms synonymously.
Response: Thank you for pointing out our lack of rigor. We have uniformly used ‘tolerance’ through the text. Please see lines 31, 93, 207, and 326.
Minor concerns
- I suggest a change in the title: BrNAC19 transcription factor acts as a positive regulator of heat stress response in Chinese cabbage
Response: many thanks for the good suggestion. We have changed the title as “Transcription factor BrNAC19 acts as a positive regulator of heat stress response in Chinese cabbage”, please see lines 3-5.
- Gene names should be written in italics. For example: BrNAC19 (lines 21-22), Superoxide Dismutase 1 (CSD1), Catalase 2 (CAT2),… Please, check the rest of the text.
Response: We have checked the whole text and italicized the gene names. Please see lines 22, 25, 51, 87, 324, 325, 326, 327.
- Species names should be written in italics and should include the author name for the first citation. For instance: Arabidopsis thaliana (L.) Heynh. (line 55).
Response: we have corrected it accordingly, please see lines 58 and 78.
- Please, review if this is correct: CUC2 (NAC) (line 58).
Response: NAM, ATAF1/2, and CUC2 (NAC) is correct, but it's still an abbreviation. In this revised version, we have changed it to its full name “No apical meristem, Arabidopsis thaliana-activating factor 1/2, and Cup-shaped cotyledon 2 (NAC). Please see lines 61-63.
- The second and subsequent times, the species names must be written with the abbreviated genus and species. For example, A. thaliana (line 90). Please check the rest of the text.
Response: We have checked the text and the corresponding species names have been revised. Please see lines 97, 100, 125, 169, 229.
- The name of the manufacturer must be accompanied by the country of manufacture. For example, lines 100, 103,…
Response: We have added this information for each manufacture, please see lines 102, 107, 109, 110, 111, 127, 142, 156, 159, and 170.
Reviewer 3 Report
Comments and Suggestions for Authors
The manuscript described the functional analysis of the transcription factor BrNAC19 from Chinese cabbage in response to heat stress. The research objective is important filling the gap of knowledge of heat stress gene regulation in Chinese cabbage. The experimental design is sound, and the data set is convincing. The discussion and conclusion are supported by the experimental data. However, primer sequences used in RT-PCR analyses are missing. They should be included so that the scientific community can repeat or utilize these sequences for confirmation or further studies.
I recommend publication with minor revisions.
Minor revision suggestions.
Change “extreme heat weather” to excessive heat events
Rephrase “molecular design” at line 36
Change “On another hand” to “On the other hand” at line 50
Italicize “Arabidopsis thaliana” throughout the manuscript
Change “, maize” to “, and maize” at line 64
Italicize “Brassica rapa L. ssp. Pekinensis” line 73
Remove “our’ at line 118, line 133
Change “heat stresses” to “heat stress” line 171
Remove “comparably” line 199
Capitalize “the” line 232.
Add a period after “replicates: line 233
Add a dash between co and transferring line 276
Comments on the Quality of English Language
The quality of English language is acceptable.
Author Response
Reviewer 3:
The manuscript described the functional analysis of the transcription factor BrNAC19 from Chinese cabbage in response to heat stress. The research objective is important filling the gap of knowledge of heat stress gene regulation in Chinese cabbage. The experimental design is sound, and the data set is convincing. The discussion and conclusion are supported by the experimental data. However, primer sequences used in RT-PCR analyses are missing. They should be included so that the scientific community can repeat or utilize these sequences for confirmation or further studies.
Response: we have listed the primers used for RT-qPCR, please see Table S1.
I recommend publication with minor revisions.
Minor revision suggestions.
Change “extreme heat weather” to excessive heat events
Response: we have replaced “excessive heat events” with “extreme heat weather”, please see lines 15 and 35.
Rephrase “molecular design” at line 36
Response: we have rephrased “molecular design” with “molecular breeding”. Please see line 38.
Change “On another hand” to “On the other hand” at line 50
Response: we have changed “On another hand” to “On the other hand” in this line, please see line 53.
Italicize “Arabidopsis thaliana” throughout the manuscript
Response: we have checked and italicized “Arabidopsis thaliana” throughout the manuscript, please see lines 58, 62, and 72.
Change “, maize” to “, and maize” at line 64
Response: we have changed “, maize” to “, and maize” in this line, please see line 68.
Italicize “Brassica rapa L. ssp. Pekinensis” line 73
Response: we have italicized “Brassica rapa L. ssp. Pekinensis” in this line.
Remove “our’ at line 118, line 133
Response: we have removed “our” in these lines. Please see lines 131, and 146.
Change “heat stresses” to “heat stress” line 171
Response: we have changed “heat stresses” to “heat stress” in line, please see line 188.
Remove “comparably” line 199
Response: we have removed “comparably” in this line, please see line please see line 216.
Capitalize “the” line 232.
Response: we have capitalized “the” in this line, please see line 251.
Add a period after “replicates: line 233
Response: we have added a period in this line, please see line 252.
Add a dash between co and transferring line 276
Response: we have added a dash in this line, please see line 296.
Round 2
Reviewer 2 Report
Comments and Suggestions for Authors
Dear authors,
Thank you for taking my previous comments into account to improve the manuscript.
The current document is much better understood and, in my view, there are no longer any doubts about its scientific rigor.
Best regards,